# In Situ Synthesis of Fe_3_O_4_ Nanoparticles and Wood Composite Properties of Three Tropical Species

**DOI:** 10.3390/ma15093394

**Published:** 2022-05-09

**Authors:** Roger Moya, Johanna Gaitán-Álvarez, Alexander Berrocal, Karla J. Merazzo

**Affiliations:** 1Instituto Tecnológico de Costa Rica, Escuela de Ingeniería Forestal, Apartado, Cartago 159-7050, Costa Rica; jgaitan@itcr.ac.cr (J.G.-Á.); aberrocal@itcr.ac.cr (A.B.); 2BCMaterials, Basque Center for Materials, Applications and Nanostructures, UPV/EHU Science Park, 48940 Leioa, Spain; karla.merazzo@bcmaterials.net

**Keywords:** wood composites, magnetic properties, tropical wood, wood modification

## Abstract

Magnetic wood is a composite material that achieves harmony between both woody and magnetic functions through the active addition of magnetic characteristics to the wood itself. In addition to showing magnetic characteristics, magnetic wood offers low specific gravity, humidity control and acoustic absorption ability. It has potential for broad applications in the fields of electromagnetic wave absorption, electromagnetic interference shielding, furniture, etc. This work reports on the synthesis of Fe_3_O_4_ nanoparticles (NPs) in wood from three tropical species (*Pinus oocarpa*, *Vochysia ferruginea* and *Vochysia guatemalensis*) using a solution of iron (III) hexahydrate and iron (II) chloride tetrahydrate with a molar ratio of 1.6:1 at a concentration of 1.2 mol/L ferric chlorate under 700 kPa pressure for 2 h. Afterward, the wood samples were impregnated with an ammonia solution with three different immersion times. The treated wood (wood composites) was evaluated for the weight gain percentage (WPG), density, ash content and Fe_3_O_4_ content by the Fourier transform infrared spectroscopy (FTIR) spectrum, X-ray diffraction (XRD) and vibrating sample magnetometry (VSM). The results show that the species *P. oocarpa* had the lowest values of WPG, and its density decreased in relation to the untreated wood, with lower ash and Fe_3_O_4_ NP content. The XRD and some FTIR signals associated with changes in the wood component showed small differences from the untreated wood. Fe_3_O_4_ NPs presented nanoparticles with the smallest diameter of (approx. 7.3 to 8.5 nm), and its saturation magnetization (M_s_) parameters were the lowest. On the other hand, *V. guatemalensis* was the species with the best Ms values, but the wood composite had the lowest density. In relation to the different immersion times, the magnetic properties were not statistically affected. Finally, the magnetization values of the studied species were lower than those of the pure Fe_3_O_4_ nanoparticles, since the species only have a certain amount of these nanoparticles (NPs), and this was reflected proportionally in the magnetization of saturation.

## 1. Introduction

Metal–organic frameworks (MOFs) have been considered to be ideal precursor candidates for microwave absorption materials (MAMs) because of their tunable structure, high porosity and large specific surface area [1,2]. Very recently, the incorporation of organic and inorganic components within the wood matrix structure has been studied with the objective of improving its properties, including biodegradation resistance and rheological, mechanical, acoustic and magnetic properties [3]. The treatment used to magnetize wood combines the properties of the matrix (wood) with the properties of the magnetic nanoparticles (NP) [4,5,6,7,8]. This new composite can be applied in different ways, such as for electromagnetic shielding, indoor electromagnetic wave absorbers and heating plates, and in heavy metal absorption [9,10,11].

Three typical methods of magnetic wood fabrication are currently known: impregnation, powder immersion and coating [9,11,12]. Among these methods, impregnation with ferrite-based NPs such as Fe_3_O_4_ [12], CoFe_2_O_4_ [13] and MnFe_2_O_4_ [14] is the most important. However, an alternative method has shown better results, which consists of treating wood with a mixture of Fe^3+^ and Fe^2+^, followed by impregnation in an ammonia solution, synthesizing in situ Fe_3_O_4_ NP through a coprecipitation chemical reaction [15,16,17].

Wood prepared by this technique exhibits new properties or enhanced responses, such as high thermal performance, electromagnetic wave absorption, magnetic attraction, improved texture, easy processing and increased dimensional stability [18,19]. In fact, the process of magnetizing the wood positively transfers the magnetic properties of the fillers (the magnetic NPs) to the new composite [20]. Therefore, magnetic treatment is a promising way to endow wood with new functions and expand the applications of wood, especially the fast-growing wood species utilized in reforestation programs [11].

Some studies have demonstrated that several methods can be applied to magnetize wood. For example, Dong et al. [11] used poplar wood from trees in fast-growing plantations and concluded that this type of treatment allows the conversion of this species to multifunctional wood with high performance that could be applied where wood-based materials with high dimensional stability are required. Wang et al. [14,21] synthesized FeNi_3_ NPs in *Pinus radiata* wood and determined that the in situ method provides soft, light and inexpensive magnetic wood composites with high industrial utility. Moreover, Gan et al. [22] used *Populus cathayana* wood and found that, despite this being a low-density species, the incorporation of Fe_3_O_4_ NPs within delignified wood provides a potential strategy to develop wood-based materials for magnetic applications.

Costa Rica has been implementing reforestation programs with fast-growing hardwoods, using a wide variety of species for wood production [23,24,25]. However, currently, the incursion into high-value markets with technologically developed products is still limited [26,27,28]. In these reforestation programs, species are harvested at early ages, which makes them juvenile wood [29] with negative properties such as low density, short fibers and the presence of growth stress [30]. 

Plantation forest trees in tropical regions represent a great production opportunity even with the aforementioned negative aspects; nevertheless, it is important to increase the quality of the wood and the added value of the products by implementing treatments that minimize the negative effects of juvenile wood [30]. Hence, studying, evaluating and developing technologies such as magnetized wood are important. 

Besides the presence of juvenile wood in young trees, tropical hardwoods present a hierarchically and chemically more complex structure than softwoods [31,32,33]. These (hardwoods) are composed of fiber cells (35–70% of all cells) with the special function of providing mechanical support, while fluid is conducted through the vessels (between 6% and 55% of the total wood cells), which are enlarged cells with thin walls and large pore spaces. 

The structural differences between softwoods and hardwoods depends on the flow of liquids within the wood, both during tree growth [34] and during the industrial harvesting process, which involves the flow of liquids inward (absorption) and outward (desorption) [35]. This makes the mechanisms of liquid flow in hardwood more complex, as more anatomical elements are involved. Treatments involving liquid flow or the introduction of NPs inside the wood matrix, especially in hardwood species, become challenging, and the wood treatments involving these species should be properly studied [26,27,36].

Given this context, the present work aimed to synthesize magnetic wood in three fast-growing species from Costa Rica, namely, *Pinus oocarpa*, *Vochysia ferruginea* and *Vochysia guatemalensis*, by in situ impregnation with FeCl_3_·6H_2_O, FeCl_2_·4H_2_O and ammonia. The study evaluated the degree of synthesis of iron NP within the wood in terms of the weight gain percentage (WGP), wood density and Fe^+3^ absorption and by emission scanning electron microscopy (SEM), Fourier transform infrared spectroscopy (FTIR), X-ray diffraction (XRD) and vibrating sample magnetometry (VSM). An understanding of this wood-improving treatment of tropical species has potential for broad applications in the fields of electromagnetic wave absorption, electromagnetic interference shielding, furniture, etc.

## 2. Materials and Methods

### 2.1. Materials

The reagents used were iron (III) hexahydrate (FeCl_3_·6H_2_O) ≥ 98% trademark purity Sigma-Aldrich (St. Louis, MO, USA); iron (II) chloride tetrahydrate (FeCl_2_·4H_2_O) ≥ 98% trademark purity Sigma-Aldrich (St. Louis, MO, USA); ammonium hydroxide solution at 30% purity of the commercial brand LABQUIMAR (San Jose, Costa Rica); absolute ethyl alcohol of the trademark J.T. Baker (Madrid, Spain); and toluene of the trademark J.T. Baker (Madrid, Spain), all distributed by Industrial Casjim Costa Rica.

Sapwood from *Pinus oocarpa*, *Vochysia ferruginea* and *Vochysia guatemalensis* wood from fast-growing forest plantations in Costa Rica was used, which has been studied and showed good liquid permeability [26] and has also been tested due to its adequate absorption of different substances in the treatments toward the improvement of its properties [26,27,36]. It is important to clarify that in this article, the magnetic permeability was not investigated; therefore, the permeability indicated in the article refers only to liquid permeability. Twenty selected sapwood boards were dried in a moisture content (MC) between 12–15%, from which samples of 6 cm long × 3 cm wide and 2 cm thick were prepared.

### 2.2. In Situ Precipitation of Iron Oxide Nanoparticles within Wood

***Samples preparation:*** The in situ preparation of iron oxide NPs in 20 sapwood samples was performed according to the procedure applied by Dong et al. [11] in which 15 samples were selected for impregnation, and 5 sapwood samples were left untreated. All samples were oven-dried at 0% MC. In this condition, the samples were placed in a 3 L capacity reactor, where a vacuum of −70 kPa was applied for 20 min. After the vacuum treatment, the sapwood samples still inside the reactor were exposed to distilled water circulating at room temperature (approximately 22 °C) to eliminate the soluble extractives in water; the process was ended when the water became clear. Afterwards, the samples were dried again to 0% MC. Next, in the same reactor, they were washed in an ethanol/toluene solution (1:2, *v*/*v*) with constant stirring at room temperature for 24 h in order to remove the extractives. Finally, the sapwood samples were dried to 0% MC.

***In situ precipitation:*** The 15 extractive-free sapwood samples were then placed back in the reactor with a solution of iron (III) hexahydrate and iron (II) chloride tetrahydrate with a molar ratio of 1.6:1 at a concentration of 1.2 mol/L ferric chloride, which corresponded to 200 g/L of iron (III) hexahydrate and 90 g/L of iron (II) chloride tetrahydrate. To prepare the solution, both solutions were mechanically mixed. For this, the samples were first placed in the reactor under a vacuum of −70 kPa for 30 min, then the solution with the iron mixture was introduced at a pressure of 700 kPa for 2 h. After this time, the 15 treated sapwood samples were removed from the reactor and divided into 3 groups of 5 samples each. Each group was tested for three different immersion times in ammonia: 12, 24 and 48 h. After the corresponding times, the specimens were washed in deionized water until reaching a neutral pH.

**Evaluation of the magnetizing process:** The magnetic material was evaluated by the weight gain percentage (WGP), absorption and density of the samples. WGP was determined from the weights of the dry samples before and after in situ precipitation, following Equation (1). The absorption of the solution was obtained by Equation (2) from the weights of the dry samples before and after in situ precipitation. The density of the samples was obtained from the weights and dimensions of the dry samples before in situ precipitation (Equation (3)).
(1)$$\mathrm{Weight}\;\mathrm{gain}\;\mathrm{percentage}\;\left(\mathrm{WGP}\right)=\frac{{\mathrm{Weight}}_{\mathrm{after}\;\mathrm{in}\;\mathrm{situ}\;\mathrm{precipitation}\;}\left(g\right)-{\;\mathrm{Weight}}_{\mathrm{before}\;\mathrm{in}\;\mathrm{situs}\;\mathrm{precipitation}}\left(g\right)}{{\mathrm{Weight}}_{\mathrm{before}\;\mathrm{in}\;\mathrm{situ}\;\mathrm{precipitation}\;}\left(g\right)}\times 100$$
(2)$$\mathrm{Solution}\;\mathrm{absortion}\;\left(\frac{L}{m^{3}}\right)=\frac{({\mathrm{Weigth}}_{\mathrm{after}\;\mathrm{in}\;\mathrm{situ}\;\mathrm{precipitation}}\left(g\right)-{\mathrm{Weigth}}_{\mathrm{before}\;\mathrm{in}\;\mathrm{situ}\;\mathrm{precipitation}\;\left(g\right)}}{\mathrm{Volume}\;\mathrm{of}\;\mathrm{sample}\;\left({\mathrm{cm}}^{3}\right)}\times \;\frac{{\left(100\;\mathrm{cm}\right)}^{3}}{1{\;m}^{3}}\times \;\frac{1\;L}{1000\;g}$$
(3)$$\mathrm{Density}\;\left(\frac{g}{{\mathrm{cm}}^{3}}\right)=\frac{{\mathrm{Weigth}}_{\mathrm{before}\;\mathrm{sample}\;\mathrm{preparation}}\left(g\right)}{{\mathrm{Volume}}_{\mathrm{before}\;\mathrm{sample}\;\mathrm{preparation}\;\left({\mathrm{cm}}^{3}\right)}}$$

The percentage of ash and the Fe_3_O_4_ NPs content formed in the treated samples were also determined. For each species and each different immersion time in ammonia and from the untreated wood, three samples were taken and ground to a size of 420 µm to 250 µm (40 and 60 mesh, respectively), and the ash percentage was determined according to ASTM-D1102-84 [37]. Once the ash content had been obtained, the ground samples were used to determine the Fe_3_O_4_ NPs content in the samples by the procedure of metal determination by direct aspiration in an atomic absorption spectrometer (AAnalyst 800, MA, USA), following ASTM D6357-19 standard [38]. Once the iron value of the tested sample had been obtained, the amount of Fe_3_O_4_ in the wood was calculated by Equation (4).
(4)$${\mathrm{Fe}}_{3}O_{4}\left(\frac{\mathrm{mg}}{\mathrm{kg}\;\mathrm{wood}}\right)=\frac{{\mathrm{Ash}\;\mathrm{content}}_{\mathrm{sample}}\left(\%\right)}{100}\times \frac{{\mathrm{Fe}}_{3}O_{4}{\;\mathrm{content}\;}_{\mathrm{sample}}\left(\mathrm{mg}\right)}{1{\;}_{\mathrm{ash}}\left(g\right)}\times \frac{1000{\;}_{\mathrm{ash}}\;\left(g\right)}{1{\;}_{\mathrm{ash}}\left(\mathrm{kg}\right)}$$
with the ash and Fe_3_O_4_ NP contents evaluated as described above.

### 2.3. Experimental Techniques

***Fourier transform infrared spectroscopy******(FTIR):*** Three samples from each species and each different immersion time in ammonia and from one untreated wood sample were ground to a size of 420 µm to 250 µm (40 and 60 mesh, respectively). The ground material was dried to 0% MC. The samples were scanned by FTIR using a Nicolet 380 FTIR spectrometer (Thermo Scientific, Mundelein, IL, USA) with a single reflecting cell (equipped with a diamond crystal). The equipment was set up to take readings by accumulating 32 scans with a resolution of 1 cm^−1^, with background correction before each measurement. The FTIR spectra obtained were processed with Spotlight 1.5.1, HyperView 3.2 and Spectrum 6.2.0 software developed by Perkin Elmer. Inc (Waltham, MA, USA). 

Field emission scanning electron microscopy: Samples with dimensions of 5 mm wide × 5 mm thick × 10 mm long from of each species and immersion time and from the untreated wood were prepared, cutting two of their corners in the form of a truncated pyramid. On the top surface of the sample, a cut was made using an American Optical Corp model 860 microtome (Búfalo, NY, USA), providing a smooth surface with a little roughness to increase the quality of the microphotography. Scanning electron microscopy (SEM) was performed with a tabletop microscope (TM 3000, Tokyo, Japan without a gold or carbon film covering the sample, using a working distance (WD) of 3.8 to 5.8 mm at 7.5 kV voltage and 400× magnification. The formation of Fe_3_O_4_ NPs and the part of the anatomical elements where the iron was deposited were observed.

***X-ray diffraction******(XRD):*** XRD was performed on samples from each species studied and each immersion time in ammonia and from untreated wood, using a PANanalytical Empyrcan Series 2 diffractometer (Worcestershire, UK), in conjunction with PANalytical High Score Plus software (version 5.1, Madrid, Spain). Sawdust from the treated and untreated samples was used and placed on a neoprene rectangle on a glass plate for measurement. The parameters of the apparatus were set as follows: Cu–Ka radiation with a graphite monochromator, a voltage of 40 kV, an electric current of 40 mA and a 2 h scan range from 5° to 55° with a scanning speed of 2°/min. The average diameter of the crystalline Fe_3_O_4_ NPs in the magnetic wood was evaluated on the basis of its XRD pattern using the Scherrer equation (Equation (5) [39], which provides a lower bound on the coherent scattering domain size, referred to as the crystallite size. The values for the equation were obtain using PANalytical High Score Plus software for the peak at 30°.
(5)$$D=\frac{K\lambda }{\left(\beta \cos\theta \right)}$$
where D is the diameter of the Fe_3_O_4_ NPs, λ is the X-ray diffraction wavelength (0.15418 nm), K is the Scherrer constant (0.89), β is the peak full width at half maximum (FWHM) and ϴ is the Bragg diffraction angle. 

***Vibrating sample magnetometry (VSM):*** The magnetic hysteresis loops (magnetization versus applied magnetic field) of wood samples of the different treatments were determined at room temperature using a MicroSense EZ7 (Milpitas, California, United States) vibrating sample magnetometer (VSM) in an external magnetic field of −20 to 20 kOe in steps of 2 Oe and 10 Oe at a low magnetic field and in steps of 100 and 500 Oe at higher fields and a time averaging of 100 ms. The saturation of magnetization (M_s_), coercivity (H_c_) and remanence (M_r_) was extracted from the hysteresis loops. The experimental percentage of the magnetic material was obtained, which was calculated by a comparison with the M_s_ (emu/gram) of the composites with the pure Fe_3_O_4_. To evaluate the stability of the magnetic properties in an acidic environment, acid resistance tests were performed by immersing the samples in a 4% hydrochloric acid solution for 7 days. The magnetic properties were then evaluated by VSM. The dimensions of the samples for the magnetic tests were 3 × 3 × 7 (longitudinal) mm^3^, and three samples were tested for each treatment and species.

### 2.4. Statistical Analysis

First, the normality and homogeneity of the data and the elimination of outliers in the variables evaluated were checked. A descriptive analysis was performed to determine the average, standard deviation and coefficient of variation for each variable measured. An analysis of variance (ANOVA) was applied with a statistical significance level of *p* < 0.05 to determine the effect of immersion time in ammonia (the independent variable) on the properties evaluated (the response variables). Tukey’s test was used to determine the statistical significance of the differences in the averages of the variables.

## 3. Results and Discussion 

### 3.1. Weight Gain Percentage, Absorption and Density of Magnetic Wood an Ash and Ferron Content

The values of WGP, absorption and the density of magnetic wood are presented in Table 1. WGP varied from 1.48% to 3.53% for *P. oocarpa*, from 4.21% to 6.49% for *V. ferruginea* and from 4.11% to 6.36% for *V. guatemalensis*. The absorption values were between 5.52 and 17.78 L/m^3^, and the density values varied from 0.20 to 0.57 g/cm^3^. 

For the magnetic wood of *Pinus oocarpa*, treatment for 12 h had the highest values of WGP, but these were similar to that of 24 h, while the sample treated for 48 h presented the lowest values of WGP (Table 1). Regarding the absorption, the sample treated for 12 h had the highest value, followed by the 24 h treatment, while the 48 h treatment presented the lowest values (Table 1). In *Vochysia ferruginea*, the highest WGP and absorption values were observed in the 12 h and 24 h treatments, and the treatment for 48 h showed the lowest values (Table 1). In *Vochysia guatemalensis*, no differences were observed among the treatments. In relation to density in the three tropical species, all magnetizing treatments decreased the wood density, for which the highest decrease occurred in *P. oocarpa* and the lowest in *V. guatemalensis* (Table 1).

As expected, ash content and Fe_3_O_4_ NP content were higher under all magnetization treatments in the three species than in the untreated wood. The ash content in *P. oocarpa*, *V. ferruginea* and *V. guatemalensis* varied between 0.15% and 4.11%, between 1.47% and 10.40% and between 2.51% and 7.85%, respectively. The highest percentage was observed in the 12 h and 24 h treatments, and the lowest values were in the 48 h treatment (Figure 1a). The Fe_3_O_4_ content varied from 0.07 to 817.19 mg/kg for *P. oocarpa*, from 0.70 to 4298.45 mg/kg for *V. ferruginea* and from 0.65 to 1907.7 mg/kg for *V. guatemalensis* (Figure 1b).

It was observed that in *P. oocarpa*, the Fe_3_O_4_ content was highest under the 12 h treatment at 817.19 mg/kg; however, no statistically significant differences were found among the three immersion times (Figure 1b). In *Vochysia ferruginea*, a statistically significant difference was observed among the different treatments: the highest content of Fe_3_O_4_ NPs was observed for 24 h of magnetization at 4298.45 mg/kg (Figure 1b). In *Vochysia guatemalensis*, the magnetization process increased the Fe_3_O_4_ NP content for all three immersion times, and the highest value was observed for the 12 h magnetization treatment at 1907.12 mg/kg (Figure 1b).

The WGP values were lower than those obtained by Dong et al. [11] for fast-growing poplar wood (*Populus tomentosa*) using the same Fe_3_O_4_ precipitation method. These authors reported an average of 27.93%, while the present study obtained values lower than 6.09% (Table 1). Meanwhile, Gao et al. [40] obtained a very wide range of WGP in fir wood samples from 5.8 to 78%; the lower limit of this range is congruent with the WGP values of the two *Vochysia* species, but *P. oocarpa* presented lower WGP values (Table 1). Likewise, the values found here are congruent with those reported by Hamaya et al. [41], who did not indicate the species used for impregnation with the iron NP but reported a WGP lower than 4.6%, a percentage slightly lower than those found in the two *Vochysia* species and similar to the percentages of *P. oocarpa* (Table 1).

Another important factor with respect to WGP and the solution absorption values is that there was a tendency for these values to decrease with an increase in the immersion time in ammonia in *P. oocarpa* and *V. ferruginea*, while the opposite occurred in *V. guatemalensis* (Table 1). This behavior may be attributed to the fact that the alkaline condition of the ammonia solution leads to partial degradation of lignin and hemicellulose [16,18,40,42,43], which accentuates the degradation of these polymers with an increasing immersion time of the wood in ammonia. This results in less precipitation of Fe_3_O_4_ NPs.

The effect of the immersion time in ammonium was confirmed by observing the values of the ash and Fe_3_O_4_ NP content. Although there was a significant increase in these parameters through the treatment for producing Fe_3_O_4_ NPs in situ, these decreased for the 48 h immersion time in *P. oocarpa* and *V. ferruginea* (Figure 1). Again, the degradation of lignin and hemicellulose that occurs when using prolonged immersion times [16,18,40,42,43] leads to reduced precipitation of Fe_3_O_4_ NPs, decreasing the amount of these (Figure 1a) and the amount of inorganic material (ash content) present in the wood after the thermal degradation process had been applied to determine ash content [41]. The increase in the amount of ash in these species agreed with studies carried out by other authors on other species [16,18,40,42,43], and they attributed the increase in inorganic material in the wood to the in situ deposition of iron.

The absorption and retention of Fe_3_O_4_ NPs varied among the different tropical hardwood species. Such variability is attributed to the wood’s permeability or the fluid flow in the wood [42], which can vary depending on the anatomical elements comprising them. In hardwood species, the liquid flow occurs mainly in the longitudinal direction through the lumina of the vessels, which are connected at the end to another vessel through the perforation plates [42]. Radial flow occurs through the radial parenchyma, which is fed by radiovascular punctuations. The liquid then flows transversally through the lumen of the ray cells, passing to other ray cells through the pits at the ends or the lateral pits of other rays [42]. Radial flow is favored when the rays comprise more than three series in width [42]. These variations that occur in these anatomical elements produce variations in the absorption (Table 1) and retention of Fe_3_O_4_ NPs (Figure 2a).

The major effect on wood after the magnetizing process was the decrease in wood density (Table 1), with the possible consequence of degradation in other wood properties due to its relationship with mechanical properties [43]. This decrease is attributed to the fact that during the deposition of the Fe_3_O_4_ NPs, the NPs attach to the inner surface of the lumen of the fiber [16], then the dissolution of lignin and hemicellulose occurs during the ammonia immersion process [44], and components with a low molecular weight are removed from the wood [41].

The decreases in density in *P. oocarpa* resulting from the Fe_3_O_4_ NP deposition method disagrees with the results for other wood magnetizing methods, such as the one developed by Oka et al. [9,18], who treated wood with Mn–Zn ferrite powder and a polyvinyl acetate resin emulsion mixed in water to create the magnetic coating material. According to Liu et al. [22], alkaline immersion makes the method more invasive in the structural components of the wood compared with other methods that do not use it.

### 3.2. SEM Observation

The SEM images showed deposits of iron NPs in different anatomical elements of the magnetic wood (Figure 2a–c) and the untreated samples of each species (Figure 2(a1,b1,c1)). In *P. oocarpa*, during the magnetizing process, the iron was deposited principally in the rays and, in a smaller amount, in the fibers (Figure 2(a2–a4)). In *V. ferruginea* (Figure 2(b2–b4)) and *V. guatemalensis* (Figure 2(c2,c3)), we can observe the iron deposits in the walls of the vessels and a smaller amount in the fibers. In addition, no wood collapse was observed due to the treatment, as occurred in *Populus* species [16] and *Fagus sylvatica* [45]. 

The process of Fe_3_O_4_ NPs deposition has been extensively detailed in different studies; however, most of them have concentrated on softwood species, such as *Pinus sylvestris* [41] and *Picea abies* [46]. Studies that have described research into the formation of Fe_3_O_4_ NPs in softwood species, which is more extensive, have focused on the location of Fe_3_O_4_ NPs within the fibers [11,17,18,47,48,49,50], leaving a void in the knowledge about the formation of NPs in other wood structures, such as vessels and rays; therefore, the present study aimed to cover this neglected element.

By observing the specific location of Fe_3_O_4_ NPs in each of the species, it was found that although *P. oocarpa* is a softwood species, the deposition of NPs mostly happened in the fibers [41,46], which was not expected, since it has been observed that in this species, deposition occurs mainly in the rays. This species has a number of resin canals in the axial and radial directions in the wood structure (over 10%), where the resin is deposited in the ducts themselves or in adjacent cells [47], thus preventing an adequate flow of liquid during the process of magnetizing the wood [48]. This situation means that the greatest amount of Fe_3_O_4_ NPs is likely to form in the rays and a little amount in the lumens of the fiber.

Fe_3_O_4_ NP formation occurs mainly in the lumina of the vessels, followed by the radial and axial parenchyma and, to a lesser extent, in the fibers (Figure 2). This pattern of in situ deposition in the different anatomical elements is attributed to the fluid transport mechanisms within the hierarchical structure [49]. The flow within hardwood happens within one of the anatomical components, the vessels [42]; therefore, greater in situ formation of Fe_3_O_4_ NPs inside this component was expected. After the vessels, the rays are the next anatomical elements in terms of the flow inside the wood, particularly for radial conduction [27]. Therefore, the formation of Fe_3_O_4_ NPs in these anatomical elements is to be expected. In the case of fibers in hardwood species, these constitute anatomical elements of trees used for structural support, not to transport liquid [42]; therefore, little in situ formation of Fe_3_O_4_ NPs is expected in the fibers (Figure 2c).

### 3.3. XRD Spectrum

The crystalline structures of untreated and treated specimens were characterized by XRD (Figure 3a–c). Untreated wood of the three species showed the typical diffraction angles at 2θ of cellulose [50] at 15.8°, 22.2° and 34.6°, corresponding to (101), (102) and (040), respectively (Figure 3a–c). The patterns of the iron composites showed that the two peaks of cellulose (at 15.8° and 22.2°) were slightly weaker in the treated specimens and that the peak at 34.6° was covered by other peaks at 35° for all three species (Figure 3a–c). According to Dong et al. [11], these changes indicate that part of the crystalline structure of the wood was damaged, confirming that the ammonia treatment affects the wood’s structure. 

In addition to these changes, magnetized wood presented diffraction peaks at 2θ: 30°, 35° and 44°, which corresponded to the (220), (311) and (400) planes of Fe_3_O4, respectively (JCPDS number 19-0629) (Figure 3a,b), indicating a cubic phase with the space of *Fd*$\bar{3}$*m* [51]. Although no differences were found among the different immersion times in ammonia, there were differences among species. In the case of the three treatments tested in *P. oocarpa*, the weakest diffraction peaks were found at 30° (220), 34° (311) and 44° (400); in fact, the peaks at 30° (220) and 34° (311) were almost null (Figure 3a). In *V. ferruginea*, the strongest peaks were located at 30° and 35°, especially under the 12 h and 24 h treatments (Figure 3b). In *V. guatemalensis*, the strongest diffraction peaks were found at 30°, 35° and 44°, especially under the 48 h treatment (Figure 3c). These differences in the patterns and the determination of particle size using the Scherrer equation (Equation (5)) [38] showed that the dimensions of the deposited Fe_3_O_4_ NPs were different in each of the species: the largest size was obtained in *V. ferruginea*, followed by *V. guatemalensis*, and the smallest was found in *P. oocarpa* (Table 2). The particle sizes in *P. oocarpa* agreed with the values found by Mashkour and Ranjbar [17]; the values for the *Vochysia* species were slightly larger than those reported by Lou et al. [16] and Gao et al. [40]. However, XRD showed inorganic particles, and we suggest that these were Fe_3_O_4_. This result must be considered with care because Fe_2_O_3_ presents a similar spectrum to Fe_3_O_4_, and the latter particle can be oxidized to Fe_2_O_3_ during storage. However, the author tried to avoid this forming during the research, and the magnetic test showed magnetic characteristics. 

The smallest NPs were found in *P. oocarpa*; the largest NPs were in *V. ferruginea,* and an intermediate size was found in *V. guatemalensis*. The time of immersion in ammonia also affected the size of the particles in the different species. The largest NPs were observed for the 48 h magnetizing process, followed by the 24 h and 12 h treatments in *P. oocarpa*. In the case of *V. ferruginea*, the largest NPs were found for the 12 h treatment, but no differences among the treatments were found in *V. guatemalensis* (Table 2). Regarding previous results about the size of NPs, *P. oocarpa* and *V. guatemalensis* presented particle sizes smaller than the values reported by Gao et al. [40], Dong et al. [11,16], Li et al. [24], and Garskaite et al. [44], but the NP size in *V. guatemalensis* was similar to the results of those studies.

### 3.4. FTIR Spectrum

FTIR was used to track the changes in the chemical composition of the untreated and treated specimens. Although in Figure 4 the range from 1950 to 3500 cm^−1^ is not presented, this region presents the absorption bands assigned to the O–H stretching vibration of hydroxyl groups (3340 cm^−1^), and the bands at 2901 corresponded to asymmetric -CH_3_ [12]. No differences were observed between the untreated and treated samples in this region. The range from 650 to 1950 cm^−1^ in the FTIR spectra showed the changes in the structures of cellulose (an alteration in the peak at 1153 cm^−1^), hemicellulose (a change in the peak at 1739 cm^−1^) and lignin (modification of the peaks at 1601, 1505, 1462, 1419 and 1251 cm^−1^). 

The main vibrations, where the greatest changes in the wood occurred, were identified at the peaks at 1251 cm^−1^, which is related to the C–O stretching vibration; 1505 cm^−1^ for the cell wall components of lignin; 1601 cm^−1^ for the cell wall components of cellulose; 1739 cm^−1^, assigned to C=O stretching in nonconjugated ketones and ester groups; 2901 cm^−1^, corresponding to asymmetric CH_3_; and 3340 cm^−1^, the absorption band assigned to the O–H stretching vibration of hydroxyl groups [11,13,22,38]. The weakened signal at 1739 ${\mathrm{cm}}^{-1}$ belonged to the C=O stretching vibration absorption peak of hemicellulose [52], which tended to be decreased by the magnetizing process in all three species. This weakening was in agreement with the studies presented by Gao et al. [40], Dong et al. [11,16], Gan et al. [13,14], Liu et al. [24], Wang, et al. [21], Yuan et al. [40] and Garskaite et al. [44]; in these studies, the weakening was attributed to hemicellulose degradation. 

The modification of the structure of cellulose was not affected: the 1145 cm^−1^ signal indicating the stretching vibration of C–O in the glucose of cellulose [39] was similar in untreated and treated wood. Meanwhile, major weakening of the lignin signal occurred at 1601 cm^−1^ (C=C aromatic skeletal vibrations [41]) and 1251 cm^−1^ (the ester linkage of carboxylic groups of the ferulic and p-coumaric acids of lignin [53]) in all three wood species; although other signs of lignin at 1505 cm^−1^, 1462 cm^−1^ and 1419 cm^−1^ did not suffer any modification.

It was seen that in *P. oocarpa*, the signal at 1739 cm^−1^ tended to decrease after the magnetizing process (Figure 4a). In *V. ferruginea*, the greatest changes in the signal intensities occurred at 1251, 1739 and 2240 cm^−1^, where the signals were weaker compared with those in the untreated samples (Figure 4b). In *V. guatemalensis*, the signals at 1251 and 1739 cm^−1^ tended to decrease after the magnetizing process, and the signal at 1601cm^−1^ increased, especially under the treatment for 48 h (Figure 4c).

According to the FTIR spectra, cellulose and hemicellulose were less affected in *V. guatemalensis*, confirming that this species was less affected by the ammonia treatment during the magnetization process. However, the lignin was moderately affected, which proves that the wood density was less affected compared with the other two species (Table 1).

The formation of Fe_3_O_4_ in the bands below 800 cm^−1^ and the bands around 561 and 667 cm^−1^ made it difficult to demonstrate the formation of Fe_3_O_4_ because there were over-laps with other signals; therefore, other techniques that measure the wavelength between 650 and 450 cm^−1^ are needed [51,54]. Despite the signal overlapping at 800 cm^−1^, it was possible to observe that this peak corresponded to tetrahedral sites of the crystal lattice, and this was a result of the oxidation of Fe^2+^ and Fe^3+^ [41], which increased mainly in the wood of *V. ferruginea* and *V. guatemalensis* (Figure 4b,c). In the wood of *P. oocarpa*, there was little evidence of a peak (Figure 4a), thus confirming the low measured Fe_3_O_4_ NP content (Figure 1a) and the poor treatment observed through SEM photographs (Figure 2a) compared with the other two wood species.

### 3.5. Magnetic Properties of Wood

Figure 5a–c show the hysteresis curves of the untreated wood and magnetic wood. The representative magnetic properties (H_c_, M_r_ and M_s_) were extracted from the hysteresis loops, while the percentage of magnetic NPs present in the samples was calculated from the saturation magnetization of the pure Fe_3_O_4_ NPs for which the value was 15.37 emu/g. As a side note, it is well known that NPs show a lower M_s_ compared with the M_s_ of their bulk materials [55]; in this case, the bulk Fe_3_O_4_ had an Ms of 93 emu/g [56].

In the case of the untreated wood, the hysteresis loops showed diamagnetic behavior with a negligible M_s_ (Figure 5), which is the expected behavior of wood. On the other hand, the treated wood (i.e., magnetic wood) of all specimens exhibited a clear hysteresis loop corresponding to ferromagnetic behavior (Figure 6). However, large differences among species were found. For example, the saturation of magnetization was the lowest (from 1.0 to 2.5 emu) in *P. oocarpa* (Figure 5a) and the highest in *V. guatemalensis* (Figure 5c), and *V. ferruginea* had intermediate values (Figure 5b). These results indicate that the species *P. oocarpa* was less magnetized and *V. guatemalensis* was the most magnetized. The variation in the saturation of magnetization of the analyzed species was produced by the variation in the precipitation of the Fe_3_O_4_ NPs, which corresponded to the different amounts of this compound that every species maintained (Figure 1a and Figure 2a); this difference in the amount is related to the variation in WGP (Table 1). It has previously been indicated that such variability is attributed to the wood’s permeability or the fluid flow in the wood [42], which varies depending on the anatomical elements of the wood. 

An important aspect regarding the values of H_c_, M_r_ and M_s_ of the species in this work is that they presented lower values compared with those of previous studies [11,40,47]. Although previous studies have used other methods of precipitating the NPs of Fe_3_O_4_, these low values may be associated with the low aptitude for the flow of liquid substances in the studied species. However, this result is congruent with other wood modification treatments in these tropical species, such as mineralization, acetylation and furfurylation, which have also shown low saturation magnetization values due to the low permeability of the liquids in the wood [27,36]. In this case, we related the differences in the magnetic properties in this work compared with other studies to the immersion times in ammonia. 

The major difference related to the M_s_ among the species and different treatment time. It is worth mentioning that this difference is not great and it lies in the differences of the corresponding experimental percentage of the Fe_3_O_4_ NPs present in the wood, as explained before, which, in some cases, increased by one order of magnitude. For example, *Pinus oocarpa* under the 24 h treatment had the highest M_s_ (Figure 5a), but still showed no statistical difference from the 12 h and 48 h treatments (Table 3). In the case of *Vochysia ferruginea*, the M_s_ values presented a linearly inverse behavior: the M_s_ decreased with an increase in immersion time. Regarding the values of H_c_, samples treated for 12 h and 48 h showed a similar hysteretic behavior between each other (i.e., similar values of H_c_); even so, these values were higher than that of the 48 h treatment (Figure 5b). In *Vochysia guatemalensis*, the treatment with the highest M_s_ was observed in the sample subjected to 12 h of the magnetizing process (Figure 5c), but there was no significant difference in the other samples with different treatment times (Table 3). According to these results, the different immersion times in ammonia had no effect on the magnetic properties of the wood, a result that disagrees with the study by Lou et al. [16], which found that the M_s_ increased with immersion time, which was attributed to the increase in the attachment of Fe_3_O_4_ generated in situ to the surface of the wood fibers.

If we compare these M_s_ values with the Fe_3_O_4_ NP content (WGP in Table 1 and Table 3), the lowest value of Ms in *P. oocarpa* agrees with the lowest NP content, although the M_s_ values of *V. guatemalensis* and *V. ferruginea* are not exactly aligned with the measured NP content. Within each species, *V. ferruginea* had a linear increase in M_s_ with an increase in NP content, as both decreased with treatment time, while in the other two species, there was no such linear relationship. This nonlinear behavior or incongruence among the species might be because there is little inhomogeneity in the dispersion of the nanoparticle content among the samples [57], resulting in small differences in the magnetic values.

In general, the values of H_c_ and M_r_ were very low (Table 3), which indicates the low magnetic properties of the wood [11]. There is also no linear tendency between the treatment time and these values (M_r_ and H_c_). One crucial consideration worth mentioning in the analysis of these values is the NP diameter found by XRD (Table 2). The mechanism by which the NPs alters the magnetization depends on their dimensions, as a maximum H_c_ can be found for the diameter at which a change between multidomain and monodomain behavior occurs, which, in the case of Fe_3_O_4_, is around 128 nm [58,59,60,61]. Without going much further into this analysis, since it is outside the scope of this work, a low H_c_ was expected, since the size of the NPs in the magnetic wood was much lower than this value (corresponding to a monodomain switching). On the other hand, superparamagnetism in Fe_3_O_4_ NPs is expected below 20 nm in diameter, for which almost no hysteresis (H_c_ ≈ 0) and lower remanence and saturation magnetization can be obtained. Since the NPs in this work had approximately these dimensions, this might contribute to these lower values. This point is under further study.

Regarding the process, previously, Gan et al. [12] indicated that the alkaline immersion is more invasive of the structural components of the wood than other magnetizing methods, so in the case of these species, the immersion time did not significantly alter the magnetic properties of the wood composites, although the immersion itself might affect the adhesion of the magnetic nanoparticles in the samples.

## 4. Conclusions

The synthesis of Fe_3_O_4_ NPs in three tropical species (*Pinus oocarpa*, *Vochysia ferruginea* and *Vochysia guatemalensis*) by in situ impregnation with Fe^3+^ and Fe^2+^ and immersion in ammonia for three different times showed different Fe_3_O_4_ nanoparticle and wood composite properties. The lowest precipitation occurred in *P. oocarpa*, resulting in lower M_s_; on the contrary, the species with the highest M_s_ was *V. guatemalensis*. In spite of the small effect on the magnetic properties, the change in wood characteristics was evident, such as a decrease in density and changes in the XDR and FTIR spectra, which are associated with changes in the cellulose, lignin and hemicellulose of the wood present in the wood composites. Contrary to other treatments in other species, the ammonia immersion time had no effect on the magnetic properties or changes in the wood structure.

## Figures and Tables

**Figure 1 materials-15-03394-f001:**
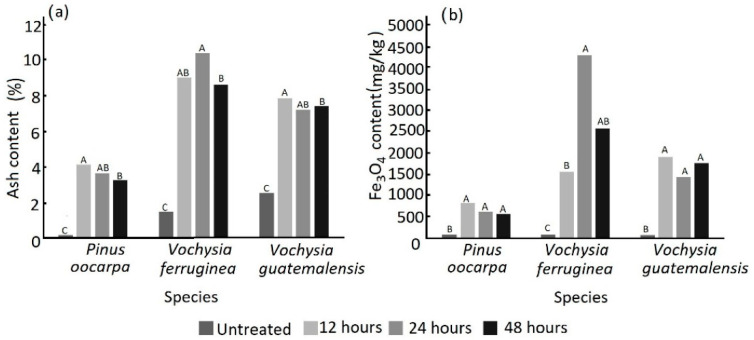
Ash (**a**) and Fe_3_O_4_ NPs (nanoparticles) (**b**) content of untreated wood and magnetic wood with three different immersion times in ammonia in three tropical wood species from fast-growth plantations in Costa Rica. Note: Letters mean significant differences between treatment at 95% in each species.

**Figure 2 materials-15-03394-f002:**
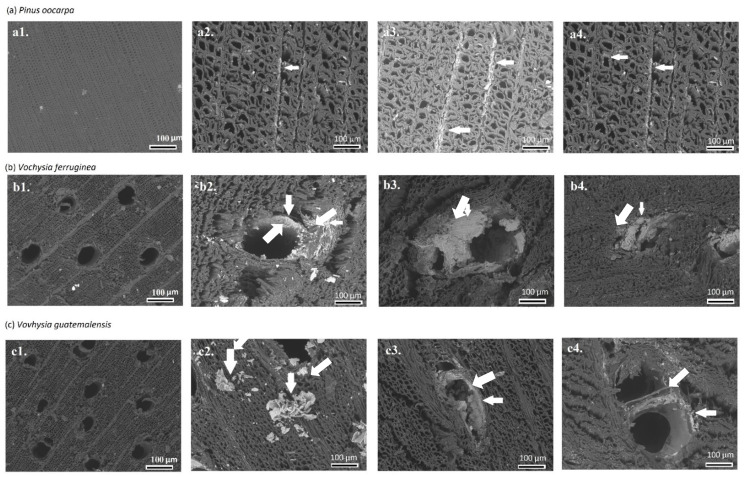
Scanning electron microscope **(**SEM) images showing Fe_3_O_4_ NPs formation of magnetic wood with three different immersion times in ammonium (1: untreated, 2: 12 h, 3: 24 h; 4: 48 h) in three tropical wood species (**a**–**c**) from fast-growth plantations in Costa Rica. Note: The white arrow indicates the formation of Fe_3_O_4_ NPs in the rays, fibers or vessels.

**Figure 3 materials-15-03394-f003:**
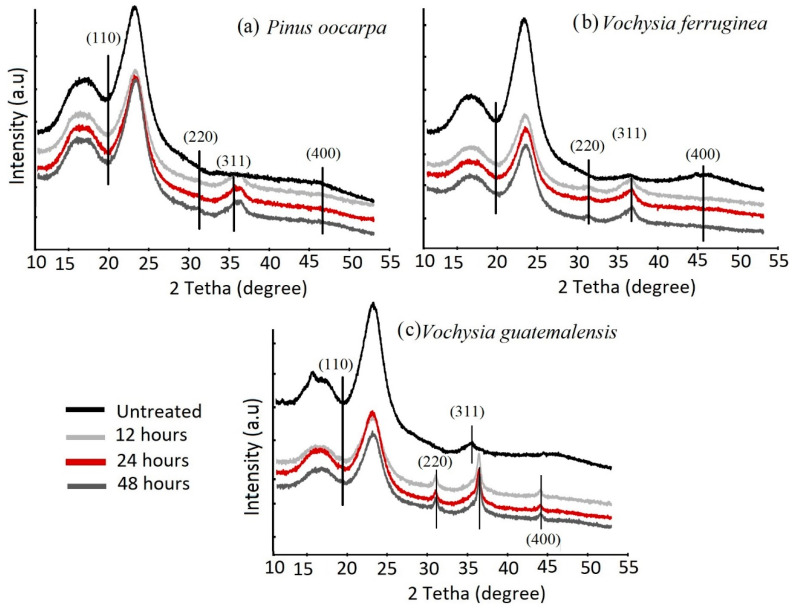
X-ray diffraction **(**XRD) in untreated wood and magnetic wood of three tropical wood species from fast-growth plantations in Costa Rica: (**a**) *Pinus oocarpa*, (**b**) *Vochysia ferruginea* and (**c**) *Vochysia guatemalensis*.

**Figure 4 materials-15-03394-f004:**
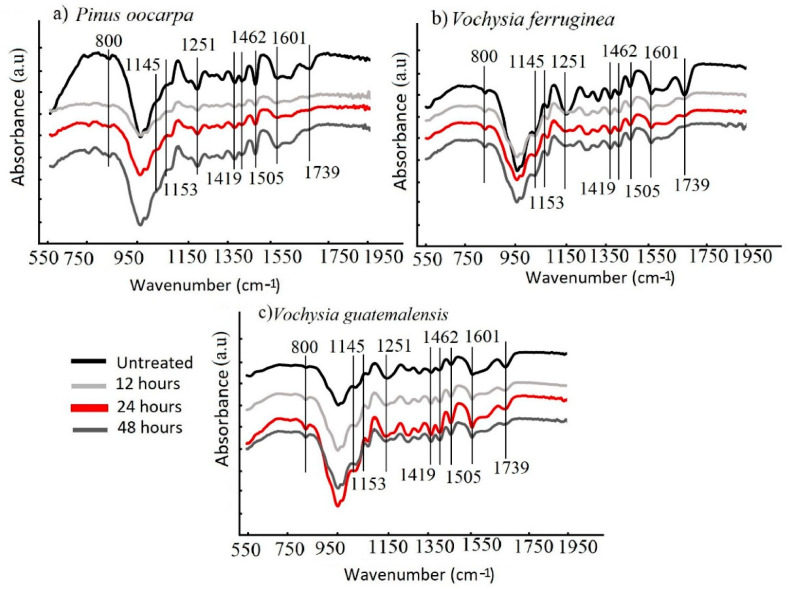
Fourier transform infrared spectroscopy **(**FTIR) spectrum of untreated and magnetic wood with three times of three tropical wood species (**a**–**c**) from fast-growth plantations in Costa Rica: (**a**) *Pinus oocarpa*, (**b**) *Vochysia ferruginea* and (**c**) *Vochysia guatemalensis*.

**Figure 5 materials-15-03394-f005:**
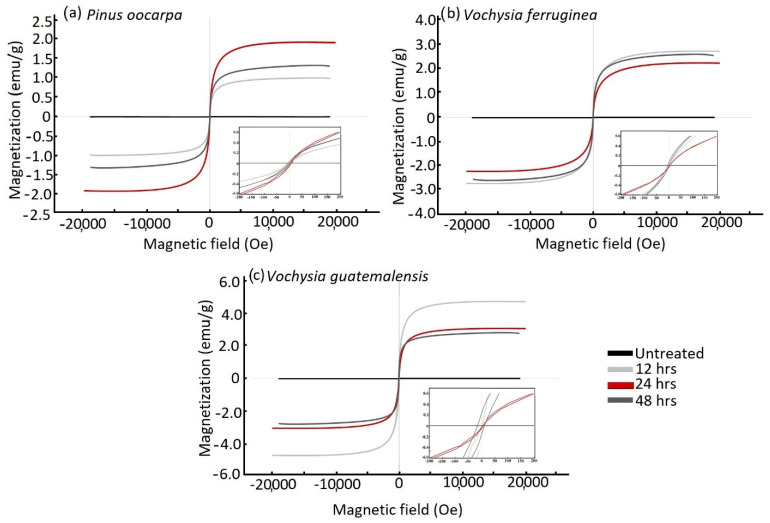
Magnetic hysteresis curves of untreated wood and magnetic wood with three different immersion times in ammonia in three tropical wood species from fast-growth plantations in Costa Rica: (**a**) *Pinus oocarpa*, (**b**) *Vochysia ferruginea* and (**c**) *Vochysia guatemalensis*.

**Figure 6 materials-15-03394-f006:**
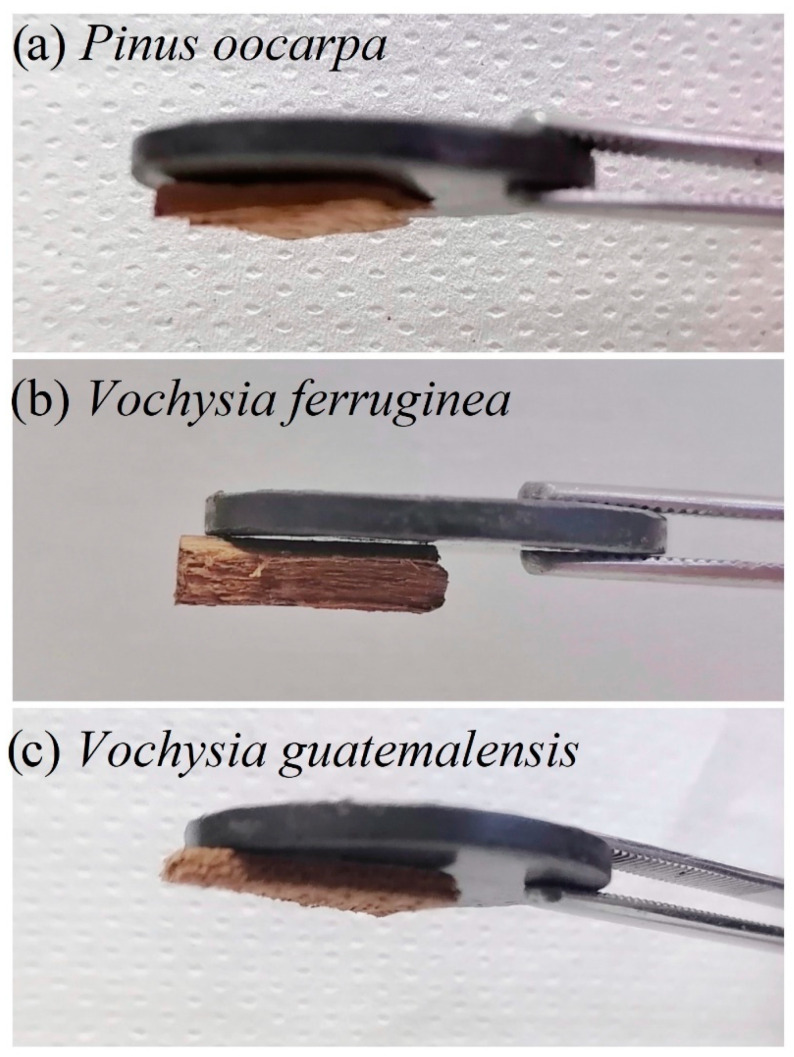
Magnetic wood samples from fast-growth plantations in Costa Rica. The figure shows a permanent magnet attracting the wood as a way to show the magnetic behavior of the samples.

**Table 1 materials-15-03394-t001:** Weight gain percentage, absorption and density of untreated wood and magnetic wood with three different immersion times in ammonia in three tropical wood species from fast-growth plantations in Costa Rica.

Species	Treatment (h)	Weight Gain (%)	Absorption (L/m^3^)	Density (g/cm^3^)
*Pinus oocarpa*	Untreated	-	-	0.57 (2.00)
	12	3.53 (55.48) ^A^	9.78 (11.70) ^A^	0.36 (4.35) ^A^
	24	2.10 (16.21) ^AB^	7.63 (13.44) ^B^	0.37 (4.29) ^A^
	48	1.48 (28.36) ^B^	5.52 (23.30) ^C^	0.38 (3.98) ^A^
*Vochysia ferruginea*	Untreated	-	-	0.38 (3.40)
	12	6.49 (13.32) ^A^	12.70 (9.48) ^A^	0.20 (3.82) ^A^
	24	6.09 (8.96) ^A^	12.00 (9.22) ^A^	0.20 (2.40) ^A^
48	4.21 (35.20) ^B^	8.30 (34.05) ^B^	0.20 (2.05) ^A^
*Vochysia guatemalensis*	Untreated	-	-	0.37 (3.01)
	12	5.38 (15.50) ^A^	14.15 (10.00) ^A^	0.27 (7.59) ^A^
	24	6.36 (48.96) ^A^	17.78 (45.12) ^A^	0.29 (4.10) ^A^
	48	4.11 (15.06) ^A^	11.07 (10.13) ^A^	0.28 (5.61) ^A^

Note: Numbers in parentheses mean coefficient of variation, and different letters in average mean significant differences between treatment at 95% in each species.

**Table 2 materials-15-03394-t002:** Particle size of Fe_3_O_4_ NPs in the magnetic wood with three different immersion times in ammonia in three tropical wood species from fast-growth plantations in Costa Rica.

Treatment (h)	Nanoparticle Size (nm)		
	*Pinus oocarpa*	*Vochysia ferruginea*	*Vochysia guatemalensis*
12	7.27 (3.13) ^B^	19.23 (1.98) ^A^	12.58 (1.78) ^A^
24	7.84 (2.15) ^AB^	18.19 (2.45) ^AB^	12.74 (2.51) ^A^
48	8.49 (2.78) ^A^	17.88 (1.96) ^B^	12.58 (2.65) ^A^

Note: Numbers in parentheses mean coefficient of variation, and letters mean significant differences between treatment at 95% in each species.

**Table 3 materials-15-03394-t003:** The coercivity (H_c_), retentivity (M_r_), saturation magnetization (M_s_) and the experimental percentage of magnetic wood with three different immersion times in ammonia in three tropical wood species from fast-growth plantations in Costa Rica.

Species	Treatment (h)	M_s_ (emu/g)	H_c_ (Oe)	M_r_ (emu/g)	Experimental Percentage (%)
*Pinus oocarta*	12	1.05 ^B^	4.39 ^A^	0.01 ^B^	6.83 ^B^
	24	1.83 ^B^	3.87 ^A^	0.01 ^B^	11.88 ^B^
	48	1.10 ^B^	3.76 ^A^	0.02 ^B^	7.16 ^B^
*Vochysia ferruginea*	12	2.98 ^B^	2.45 ^A^	0.12 ^B^	19.40 ^B^
	24	2.34 ^B^	0.73 ^A^	0.00 ^B^	15.22 ^B^
	48	2.21 ^B^	2.34 ^A^	0.02 ^B^	9.56 ^B^
*Vochysia guatamalensis*	12	3.74 ^B^	11.87 ^A^	0.17 ^A^	24.35 ^B^
	24	3.05 ^B^	7.08 ^A^	0.12 ^A^	19.87 ^B^
	48	3.29 ^B^	17.73 ^A^	0.25 ^A^	21.38 ^B^

Note: Letters mean significant differences between treatment at 95% in each species.

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
