# Peer review of "In Situ Synthesis of Fe3O4 Nanoparticles and Wood Composite Properties of Three Tropical Species"

_materials, 2022, doi:10.3390/ma15093394_

Round 1

Reviewer 1 Report

This manuscript can be accepted as it is now.

Author Response

This manuscript can be accepted as it is now.

Author answer: Thank a lot for this comment.

Here file with changes indicated for others reviewers

Reviewer 2 Report

This work reported the magnetic properties for of three species tropical wood prepared in fast growth plantation. The authors obtained magnetic wood by soaking the wood in a solution containing iron ions, and studied their magnetic characteristics and the reason for magnetism of wood. The formatting of the manuscript has been greatly improved and references are cited correctly, the conclusions are consistent with the evidence and arguments presented. The text is clear and easy to read, but some labeling errors need attention.

The manuscript has been improved greatly, but some errors still exist, such as the statement after equation (5) is wrong,D should be  Scherrer constant, λ should be e X-ray diffraction wavelength. The markup of many parameters is not standardized,such as “values of Hc, Mr and Ms” in the conclusion part. The authors should  check text formatting carefully.

Author Response

(x) English language and style are fine/minor spell check required
Author answer: the English correction was improved

The text is clear and easy to read,

Author answer: Thank a lot for this comment.

The manuscript has been improved greatly, but some errors still exist, such as the statement after equation (5) is wrong,D should be  Scherrer constant, λ should be e X-ray diffraction wavelength. The markup of many parameters is not standardized,such as “values of Hc, Mr and Ms” in the conclusion part. The authors should  check text formatting carefully.

Author answer: The equation (5) was clarified, and all abbreviations were standardized across the text

Here file with changes indicated for other reviewers

Reviewer 3 Report

The manuscript titled “Tropical magnetic wood properties prepared in situ from iron oxide nanoparticles of three species growing in fast growth plantation” by Moya et al. investigates the structural and magnetic properties of wood-based samples (of three different tropical species) embedded with Fe3O4 nanoparticles. The wood species were treated by impregnation of Fe3+ and Fe2+ and immersion in ammonium for different times (12, 24 and 48h). After treatment, the wood samples were studied by different techniques such as Fourier-transform infrared spectroscopy (FTIR) spectrum, X-Ray Diffraction (XRD) and vibrating sample magnetometry (VSM). None of the techniques used seems to provide compelling evidence of successful nanoparticle precipitation, starting from the magnetic measurements showing no hysteresis loop and negligible residual magnetization. Beyond the scarce scientific advancement, the manuscript lack a proper presentation and scientific discussion.

  • The abstract lack a description of the scientific context and the general interest in the topic. Also the first sentence of the Abstract “This work reports the synthetization of Fe3O4 nanoparticles using Fe3+ and Fe2+, followed by an impregnation with an ammonia solution in 3 immersion times into wood from tropical species (Pinus oocarpa, Vochysia ferruginea and Vochysia guatemalensis) to tailor the magnetic response of the resulting composites.” does not provide a sufficiently clear short description of the nanoparticle synthesis and sample treatment. Later it provides some too technical information about the results (e.g. “…signals associated with changes in the wood component (1153, 1739, 1601, 1505, 1462, 1419 and 1251 cm-1) showed no difference with untreated wood,…” and at the and it does not provide any clear conclusion or perspective.
  • In the manuscript, experimental data (e.g. in Table 1 ) do not present error bars.
  • The captions do not provide a clear description of the Figures.
  • Neither the text nor Figure 2 clearly evidence where the presence of magnetic nanoparticles is expected (Figures could be improved by placing arrows or mark points). Further, it is not clear if the reported images are sufficiently representative of the sample morphology.
  • In the XRD spectra, the authors state that magnetic wood samples present strong diffraction peaks (only poorly marked in Figure 3) that, however, seem to not match the signals of the reported spectra. A better explanation of the results must be provided. Beyond that, curves in Fig. 3 could be presented in stacked mode to improve visibility.
  • The author should discuss how the nanoparticle size of Table 2 is derived, which is the statistics, and provide an experimental error.
  • Similarly to XRD, FTIR spectra are not properly described. It is hard for the reader to distinguish, in the Figure, and understand, from the text, between the signals expected from magnetic nanoparticles and those detected in the spectra.
  • The magnetic properties reported in Figure 5 and labelled as “magnetic hysteresis loops” do not report any evidence of opened hysteresis. Indeed Table 3 reports no remnant magnetization. Further, it is not clear how the emu/g values are derived for each sample. This is a major issue for correctly comparing magnetization values of the samples.
  • It is not very clear the meaning of Figure 6.

For these reasons, although the topic could be of a general interest to the readers of Materials, I do not believe the manuscript should be published in the present form.

Author Response

English language and style, (x) Extensive editing of English language and style required
Author answer: Although this review recommended extensive English editing, this observation was not considered because Other four reviewers were not considered this editing, all recommended “fine/minor spell check required”

 Comments and Suggestions for Authors

None of the techniques used seems to provide compelling evidence of successful nanoparticle precipitation, starting from the magnetic measurements showing no hysteresis loop and negligible residual magnetization. Beyond the scarce scientific advancement, the manuscript lack a proper presentation and scientific discussion.

Author answer: The article was review for other 4 reviewer and anyone indicated any observations about lack evidence of successful, which the authors agreed with this observation.

  • The abstract lack a description of the scientific context and the general interest in the topic.

Author answer: It was added one line with following information “The incorporation of organic and inorganic components within the wood matrix structure has been studied with the objective to improve its properties”

  • Also the first sentence of the Abstract “This work reports the synthetization of Fe3O4 nanoparticles using Fe3+ and Fe2+, followed by an impregnation with an ammonia solution in 3 immersion times into wood from tropical species (Pinus oocarpa, Vochysia ferruginea and Vochysia guatemalensis) to tailor the magnetic response of the resulting composites.” does not provide a sufficiently clear short description of the nanoparticle synthesis and sample treatment.

Author answer: This sentence was re-written accruing “This work reports the synthetization of Fe3O4 nanoparticles in wood samples un-extractives using a solution of Iron (III) hexahydrate and iron (II) chloride tetrahydrate with a molar ratio of 1.6:1 at a concentration of 1.2 mol/L ferric chloat a pressure of 700 kPa for 2 hours, Followed by an impregnation with an ammonia solution in 3 immersion times into wood from tropical species….”

  • Later it provides some too technical information about the results (e.g. “…signals associated with changes in the wood component (1153, 1739, 1601, 1505, 1462, 1419 and 1251 cm-1) showed no difference with untreated wood,…” and at the and it does not provide any clear conclusion or perspective.

Author answer: Thank a lot for this comment. This part was rewritten.

  • In the manuscript, experimental data (e.g. in Table 1 ) do not present error bars.

Author answer: There is mistakes, The value in parenthesis presents variation coefficient

  • The captions do not provide a clear description of the Figures.

Author answer: All captions of figures and tables was improved.

  • Neither the text nor Figure 2 clearly evidence where the presence of magnetic nanoparticles is expected (Figures could be improved by placing arrows or mark points). Further, it is not clear if the reported images are sufficiently representative of the sample morphology. Author answer:

Author answer: Thank a lot for this comment. This figure was improved, white arrow indicates the NPs formation.

  • In the XRD spectra, the authors state that magnetic wood samples present strong diffraction peaks (only poorly marked in Figure 3) that, however, seem to not match the signals of the reported spectra. A better explanation of the results must be provided. Beyond that, curves in Fig. 3 could be presented in stacked mode to improve visibility.

Author answer: Thank a lot for this comment. The figure 3 was improved and the text was written according to the figure

  • The author should discuss how the nanoparticle size of Table 2 is derived, which is the statistics, and provide an experimental error. Author answer:

Author answer: Thank a lot for this comment. the information was added the text was written according to table.

  • Similarly to XRD, FTIR spectra are not properly described. It is hard for the reader to distinguish, in the Figure, and understand, from the text, between the signals expected from magnetic nanoparticles and those detected in the spectra. Author answer: The author disagreed with this observation. The figure as was presented, It can observed the differences.

  • The magnetic properties reported in Figure 5 and labelled as “magnetic hysteresis loops” do not report any evidence of opened hysteresis. Indeed Table 3 reports no remnant magnetization. Further, it is not clear how the emu/g values are derived for each sample. This is a major issue for correctly comparing magnetization values of the samples.

Author answer:

Thank you for the comment, although we do not agree with the referee. The so-labelled “magnetic hysteresis loops” or “Magnetic hysteresis curves” is referring to the measurements performed to the samples from the VSM and that show the magnetic properties of the wood samples, although they show low hysteresis (Hc), they are magnetic.

As it is well known [[i],[ii]] that the magnetic behaviour of magnetic-nanoparticles is not the same as the bulk material, due to different contributions that depend on the low-dimensions and the shape of the nanoparticles. The magnetic regime they are within is dictated by the dimension; the superparamagnetic regime lies at very low dimensions (diameters smaller than the threshold diameter - Dt), after this Dt goes to a single-domain regime, and then to a multi-domain one after the critical diameter (Dc) [[iii],[iv]].

In the case of this work, the presented diameter of nanoparticles is between 7. 3 and 8.5 nm, and this fact is explained in the manuscript: “On the other hand, superparamagnetism in Fe3O4 nanoparticles is expected under 20 nm of diameter, where almost no hysteresis (Hc ≈ 0), and lower remanence and saturation magnetization are obtained. Since the NP of this work are around these dimensions, this might contribute to these lower values.” In this way, we have proven that the nanoparticles present superparamagnetic behaviour.

The samples are a mix of wood and Fe3O4, and the amount of the later is not very high due to the process. This is also explained along the manuscript. The emu/g that are shown in the graphs is the total weight of the samples, i.e. wood + Fe3O4-NP. This carry an inherent error, but thanks to the Ms values, we can estimate the amount to magnetic particles within the wood. In the text we described as followed: “Also, the experimental percentage of the magnetic material were obtained, which is calculated by comparing the Ms (emu/gram) of the composites with the pure Fe3O4.”

  • It is not very clear the meaning of Figure 6.

Author answer: Thanks for this comment, we have added to the figure caption the next text: “The figure shows a permanent magnet attracting the wood as a way to show the magnetic behaviour of the samples”.

Here file with changes indicated for other reviewers

[i] Costica Caizer, Nanoparticle Size Effect on Some Magnetic Properties. Springer International Publishing Switzerland 2016 M. Aliofkhazraei (ed.), Handbook of Nanoparticles, Page 475.

[ii] Robert C. O’Handley. Modern Magnetic Materials, Principles and Applications. John Wiley & Sons, Inc. 2000

[iii] Costica Caizer, Nanoparticle Size Effect on Some Magnetic Properties. Springer International Publishing Switzerland 2016 M. Aliofkhazraei (ed.), Handbook of Nanoparticles, Page 475.

[iv] Nicola Spaldin. Book: Magnetic Materials, Fundamentals and Device Applications”. Cambridge University Press

Review 5

It is an excellent written and presented paper with a high degree of novelty. Such studies dealing with the magnetic wood properties are very very limited in the literature. Especially the application of iron oxide nanoparticles increases the novelty of the paper. I really enjoy this paper and I recommend publication in its current form. Well done to authors 

Author answer: Thank a lot for this comment.

Here file with changes indicated for other reviewers

____________

[1] Costica Caizer, Nanoparticle Size Effect on Some Magnetic Properties. Springer International Publishing Switzerland 2016 M. Aliofkhazraei (ed.), Handbook of Nanoparticles, Page 475.

[1] Robert C. O’Handley. Modern Magnetic Materials, Principles and Applications. John Wiley & Sons, Inc. 2000

[1] Costica Caizer, Nanoparticle Size Effect on Some Magnetic Properties. Springer International Publishing Switzerland 2016 M. Aliofkhazraei (ed.), Handbook of Nanoparticles, Page 475.

[1] Nicola Spaldin. Book: Magnetic Materials, Fundamentals and Device Applications”. Cambridge University Press

Reviewer 4 Report

-The current work focuses on Tropical magnetic wood properties prepared in situ from iron oxide nanoparticles of three species growing in fast-growth plantations. The author’s some effort into the manuscript, but major issues should be addressed.

Extensive editing of the English language and style is required. The lengthy sentences should be split into smaller sentences without a change in their meaning. The results are not clearly presented. The main problem in the manuscript is that the authors show only results without any interpretations of it or confirmation by citation. More details are required to explain the obtained results. Also, it should clear the author's contribution in comparison to other previous works during the discussion.

Title

The title is too long and not very suggestive

Abstract

The lengthy sentences should be split into smaller sentences without a change in their meaning.

e.g. “The present work aims to fabricate magnetic wood via In-situ precipitation technique.”

Introduction:

-  The introduction does not provide sufficient background, where all relevant references are not included.

- At the end of the introduction, the novelty of this work is not highlighted and it should clear the author's contribution in comparison to other previous works.

Experimental Details 

-              Wight by gm plus number of moles should be inserted for easier reproducible by the reader

-              In-situ precipitation of iron oxide nanoparticles within wood: No information about how mixed? Mechanical or magnetic stirrer, speed?

-              This paragraph belongs to results and discussion not to the experimental part “The main vibrations, where the greatest changes in the wood occurred, were identified at the peak 1251 cm-1 ………..O-H stretching vibration of hydroxyl groups [11,13,22,38].”

Results and discussion

-              EDX and elemental mapping are required to control the distribution and metal content

-              Figure 1 what do you mean by A, B, AB, C?????

-              In figure 2, SEM picture for net wood should be inserted to clear compared with magnetic wood

-              XRD diffraction interpretation in the discussion section is not informative and it’s general. Please rephrase this section.

-              Figure 3, please redraw it with a better resolution

-              Although the Authors claim to make scans from 5° to 90°,  but where indexed peaks for (511) and (440) for the magnetite phase!!!!

-              The indexed position is not completely satisfied!!!

Fig.3a Pinus oocarpano, NO indexed peak appears for (111), (220), (400) but Authors claim it

Fig.3b Vochysia ferruginea, NO indexed peak appears for (111), (400) but Authors claim it

-              TGA analysis is required to control the thermal stability of the sample

-              Figure 4, please redraw it with a better resolution

-              Zoom in the region between 500 and 800cm-1 and also Fe-O cannot be confirmed in this region!

-              Figure 4b where left legend? and What is the absorbance unit

-              From xrd and FTIR analysis, am not sure the type of nanoparticle phase is magnetite!!!

-              Figure 5, should insert zoom in the picture to clear coercivity (Hc), remanence (Mr) values

-              Please correct 15,37 emu/g to 15.37 emu/g

-              Figure 6 make zooms out to clear attraction to the magnet and also name the samples in the figure

-              The authors show only results without any interpretations of it or confirmation by citation. More details are required to explain the obtained results. Also, it should clear the author's contribution in comparison to other previous works during the discussion.

-  The conclusion should rephrase to the target with a specific outcome, not with the general meaning  

References

-Most of the references are not updated, please include recent references

Author Response

It is an excellent written and presented paper with a high degree of novelty. Such studies dealing with the magnetic wood properties are very very limited in the literature. Especially the application of iron oxide nanoparticles increases the novelty of the paper. I really enjoy this paper and I recommend publication in its current form. Well done to authors

Author answer: Thank a lot for this comment.

Here file with changes indicated for other reviewers

Reviewer 5 Report

It is an excellent written and presented paper with a high degree of novelty. Such studies dealing with the magnetic wood properties are very very limited in the literature. Especially the application of iron oxide nanoparticles increases the novelty of the paper. 

I really enjoy this paper and I recommend publication in its current form. Well done to authors 

Author Response

(The authors gave the same response as above.)

Round 2

Reviewer 3 Report

The manuscript has been barely improved after revisions. I still believe that the abstract should be modified to provide a general overview of the context and perspectives of the topic before providing a synthetic description of the obtained results. Figure 2 has been magnified, but it does not highlight  which diffraction peaks are expected on the untreated wood and those  expected on the treated ones. Peaks related to the planes (110), (220) and (400) coming from magnetite should be visible in principle just on the treated  woods. Apparently, this is not the case. The only appreciable difference between treated and untreated wood is related to the (311) plane signal. However, it also appears on the untreated wood in Fig 3b and it is shifted after treatment in Fig 3c   (if I understand correctly marks in Fig 3c). All these issues must be highlighted and clearly explained in the text. Similar considerations can be drawn for Figure 4, as already stated in my previous revision. In this case, the Figure has been not even barely improved and marks or labels are still poorly visible. The authors' answer to my previous request for improvement "Similarly to XRD, FTIR spectra are not properly described. It is hard for the reader to distinguish, in the Figure, and understand, from the text, between the signals expected from magnetic nanoparticles and those detected in the spectra." was "The author disagreed with this observation. The figure as was presented, It can observed the differences.". This sentence does not answer my comment and it is not even grammatically correct.

I think that resolving the above-mentioned issues could significantly improve the readability of the manuscript given the potential great interest in the treated topic. Therefore I think that they should be addressed before publication. 

Author Response

See attach file

Reviewer 4 Report

-The current work focuses on Tropical magnetic wood properties prepared in situ from iron oxide nanoparticles of three species growing in fast-growth plantations. The author’s some effort into the manuscript, but major issues should be addressed.

Extensive editing of the English language and style is required. The lengthy sentences should be split into smaller sentences without a change in their meaning. The results are not clearly presented. The main problem in the manuscript is that the authors show only results without any interpretations of it or confirmation by citation. More details are required to explain the obtained results. Also, it should clear the author's contribution in comparison to other previous works during the discussion.

Title

The title is too long and not very suggestive

Abstract

The lengthy sentences should be split into smaller sentences without a change in their meaning.

e.g. “The present work aims to fabricate magnetic wood via In-situ precipitation technique.”

Introduction:

-  The introduction does not provide sufficient background, where all relevant references are not included.

- At the end of the introduction, the novelty of this work is not highlighted and it should clear the author's contribution in comparison to other previous works.

Experimental Details 

-              Wight by gm plus number of moles should be inserted for easier reproducible by the reader

-              In-situ precipitation of iron oxide nanoparticles within wood: No information about how mixed? Mechanical or magnetic stirrer, speed?

-              This paragraph belongs to results and discussion not to the experimental part “The main vibrations, where the greatest changes in the wood occurred, were identified at the peak 1251 cm-1 ………..O-H stretching vibration of hydroxyl groups [11,13,22,38].”

Results and discussion

-              EDX and elemental mapping are required to control the distribution and metal content

-              Figure 1 what do you mean by A, B, AB, C?????

-              In figure 2, SEM picture for net wood should be inserted to clear compared with magnetic wood

-              XRD diffraction interpretation in the discussion section is not informative and it’s general. Please rephrase this section.

-              Figure 3, please redraw it with a better resolution

-              Although the Authors claim to make scan from 5° to 90°,  but where indexed peak for (511) and (440) for magnetite phase!!!!

-              The indexed position is not completely satisfied!!!

Fig.3a Pinus oocarpano, NO indexed peak appears for (111), (220), (400) but Authors claim it

Fig.3b Vochysia ferruginea, NO indexed peak appear for (111), (400) but Authors claim it

-              TGA analysis is required to control the thermal stability of the sample

-              Figure 4, please redraw it with a better resolution

-              Zoom in the region between 500 and 800cm-1 and also Fe-O cannot be confirmed in this region!

-              Figure 4b where left legend? and What is the absorbance unit

-              From xrd and FTIR analysis, am not sure the type of nanoparticle phase is magnetite!!!

-              Figure 5, should insert zoom in the picture to clear coercivity (Hc), remanence (Mr) values

-              Please correct 15,37 emu/g to 15.37 emu/g

-              Figure 6 make zooms out to clear attraction to the magnet and also name the samples in the figure

-              The authors show only results without any interpretations of it or confirmation by citation. More details are required to explain the obtained results. Also, it should clear the author's contribution in comparison to other previous works during the discussion.

-  The conclusion should rephrase to the target with a specific outcome, not with the general meaning  

References

-Most of the references are not updated, please include recent references

Author Response

see attach file

Round 3

Reviewer 4 Report

All issues are solved and the manuscript is accepted in the present form 

This manuscript is a resubmission of an earlier submission. The following is a list of the peer review reports and author responses from that submission.

Round 1

Reviewer 1 Report

The authors investigated the magnetic properties of different woods by  an impregnation with an ammonia solution. The experimental section is detailed, but there are problems with many test results that need to be improved.

  1. There are many spelling errors in the manuscript,such as:"where K is the X-ray wavelength (0.15418 nm)"in page 5; " Figure 3. XDR diffraction of magnetic wood" in page 9;
  2. In some of the figures, the author uses different colors to distinguish the different results, but the display is very bad.
  3. The sample size used in the magnetic test is too large and the results do not take into account the effect of the shape factor on the results.
  4. The table in the manuscript does not conform to the format.
  5. The distribution of Fe elements in wood can be analyzed by EDX, which can better support the conclusion
  6. References should be selected within five years as far as possible

Reviewer 2 Report

This manuscript describes the formation of magnetic wood with wood resources with three different wood species. Also the modification efficiency based on immersion time of amino alkaline solution is discussed. The authors get good magnetic wood samples, and their results partially support their conclusions. However, revisions are suggested before acceptance. My comments are as below:

  1. Before the XRD results, the authors can not confirm that the inorganic particles are Fe3O4. The authors should correct the corresponding statements.
  2. From the XRD results, the authors can not confirm that the inorganic particles are Fe3O4. This is because that the XRD curves of Fe2O3 should be similar to that of Fe3O4. And the Fe3O4 particles can be oxidized to Fe2O3 during storage.
  3. The authors should tell us in details how they can get the size information of the particles.
  4. What’s the meaning of experimental percentage in Table 3?
  5. In Figure 6, should the black sample is magnetic wood? The authors should label it.
  6. Some spelling mistakes should be corrected, For example, Fe3O4 should subscript.
  7. Some recent publications should be cited to show a background introduction of keeping pace with the times. For example, Composites Part B: Engineering, 2021, 226,109335; Nano-Micro Letters,2021, 14(1):11; et al.